# Improvement of Mucosal Lesion Diagnosis with Machine Learning Based on Medical and Semiological Data: An Observational Study

**DOI:** 10.3390/jcm11216596

**Published:** 2022-11-07

**Authors:** Antoine Dubuc, Anissa Zitouni, Charlotte Thomas, Philippe Kémoun, Sarah Cousty, Paul Monsarrat, Sara Laurencin

**Affiliations:** 1School of Dental Medicine and CHU de Toulouse—Toulouse Institute of Oral Medicine and Science, 31062 Toulouse, France; 2Center for Epidemiology and Research in POPulation Health (CERPOP), UMR 1295, Paul Sabatier University, 31062 Toulouse, France; 3Oral Surgery and Oral Medicine Department, CHU Limoges, 87000 Limoges, France; 4InCOMM, I2MC, UMR 1297, Paul Sabatier University, 31062 Toulouse, France; 5RESTORE Research Center, Université de Toulouse, INSERM, CNRS, EFS, ENVT, Université P. Sabatier, CHU de Toulouse, 31300 Toulouse, France; 6LAPLACE, UMR 5213 CNRS, Paul Sabatier University, 31062 Toulouse, France; 7Artificial and Natural Intelligence Toulouse Institute ANITI, 31013 Toulouse, France

**Keywords:** artificial intelligence, oral pathology, oral dermatology

## Abstract

Despite artificial intelligence used in skin dermatology diagnosis is booming, application in oral pathology remains to be developed. Early diagnosis and therefore early management, remain key points in the successful management of oral mucosa cancers. The objective was to develop and evaluate a machine learning algorithm that allows the prediction of oral mucosa lesions diagnosis. This cohort study included patients followed between January 2015 and December 2020 in the oral mucosal pathology consultation of the Toulouse University Hospital. Photographs and demographic and medical data were collected from each patient to constitute clinical cases. A machine learning model was then developed and optimized and compared to 5 models classically used in the field. A total of 299 patients representing 1242 records of oral mucosa lesions were used to train and evaluate machine learning models. Our model reached a mean accuracy of 0.84 for diagnostic prediction. The specificity and sensitivity range from 0.89 to 1.00 and 0.72 to 0.92, respectively. The other models were proven to be less efficient in performing this task. These results suggest the utility of machine learning-based tools in diagnosing oral mucosal lesions with high accuracy. Moreover, the results of this study confirm that the consideration of clinical data and medical history, in addition to the lesion itself, appears to play an important role.

## 1. Introduction

In the past 30 years, there has been a major increase in interest in Artificial Intelligence (AI) in medicine. Indeed, this craze is explained by the fact that practitioners need help to accelerate treatment, to have a predictive attitude, or to develop telemedicine tools. Recently, this interest has reached the field of dermatopathology [1,2], with over 600 publications referenced in Pubmed covering this subject and its different applications based on classification, segmentation, or feature extraction of lesions [3] and even smart-phone based images [4]. Although these applications are currently limited to a few types of lesions, such as skin cancers (particularly melanoma [5,6,7]), oral squamous cell carcinomas [4,8] (OSCC), ulcers, psoriasis, and atopic dermatitis [9], it has been suggested that the accuracy of AI-based diagnosis using convolutional neural networks for lesion image analysis may be similar to experienced dermatologists [10], or even better [11]. Machine Learning (ML) can be defined as a branch of AI whose algorithms learn autonomously to make predictions from data and improve their performance over time [9]. ML corresponds to a wide range of algorithm families and complexities, from self-explainable models (e.g., linear regression, polynomial regression) to other models requiring the development of additional explainability methods. (e.g., support vector machine, tree-based models, neural networks). ML models are particularly well suited to capture the complex relationships between variables to extract hidden patterns in the datasets to respond as efficiently as possible to the prediction to be made. Often referred to as “black-boxes” models, the question of how humans can understand the determinants of the prediction is crucial, particularly in the area of healthcare. Explainability is also more than a technological problem, it involves among other ethical, societal, and legal issues [12]. They are therefore of major interest to support the practitioner referring the patient to a specialist (telemedicine), but also in the diagnostic process and in guiding them towards the most appropriate treatment. The practitioner must be able to explain to the patient how the algorithm works and the criteria for the decision process [13]. Although the publications on skin lesions concern most of the work, the contribution of AI in the field of oral mucosa pathology is relatively recent [4,8,14,15]. However, the early diagnosis and management of oral mucosa lesions is a major issue, particularly in the context of oral cancer diagnosis and management. According to the World Health Organization (WHO), lip and oral cavity cancers are the 16th most frequent cancers, corresponding to more than 177,000 deaths worldwide in 2020. Oral cancer and primarily squamous cell carcinoma accounted for half of the overall annual mortality from head and neck cancer in 2018 [16]. Since 2019, some teams have been interested in the contribution of AI algorithms and especially ML in the management of oropharyngeal cancer, mainly in terms of prognosis prediction [17,18,19] and medical image classification [20], but also in optimizing the treatment of these cancers [21]. However, the lack of practitioners specialized in oral mucosa pathology and the variety of lesions have made differential diagnosis a major challenge and the goal of improving early diagnosis and patient management.

This study aimed to propose a predictive ML-based framework with state-of-the-art explainability techniques based on chair-side patient information collection, such as the semiology of the lesions and patient socio-demographic and medical data for diagnosis of oral mucosa lesions.

## 2. Materials and Methods

### 2.1. Selection Criteria

Patients consulting in the specialized oral dermatology consultation of the Toulouse University Hospital (CHU de Toulouse) between January 2015 and December 2020 were considered in this study.

All patients meeting the following criteria were included in the study: (1) Patients consulting in the oral mucosa pathology consultation of the Toulouse University Hospital and presenting a mucosal lesion, (2) Patients already followed for such lesions and requiring a histological re-evaluation, (3) Subjects older than 18 years of age at the time of consultation, (4) Subjects who are able to understand French language, to receive information about the study and understand the information form to participate in the study, (5) Subjects who have given their consent to participate in the study. Patients with the following conditions could not be included in this study: (1) Pregnant and lactating women, (2) Patients under guardianship, curatorship, or safeguard of justice. All patients gave their consent for data collection and the computer processing of personal and medical data was implemented to the results of this research. Patients of legal age were included in the study, patients over 18 years of age.

This study was conducted in accordance with the public health code and bioethics laws governing research in France [22]. This study is retrospective and does not involve the human person. Therefore, it is not submitted to an ethics committee but complies with the reference methodology (MR-004 of the French National Commission for Informatics and Liberties (CNIL) (number: 2206723 v0)) for which Toulouse University Hospital signed a commitment of compliance. The patient’s non-objection was collected.

### 2.2. Data Collection and Annotation

For each subject, a series of high-resolution, macroscopic photographs of the lesion (either by digital camera or smartphone) were acquired. When for the same patient, several images of the same lesion were taken, only one of them was considered, whereas images of the same patient but representing a different lesion and/or a different representation of the same lesion (e.g., according to topography or color) were treated separately. Each lesion characteristic (type of lesion, color, topography, size, homogeneity, and shape) was annotated by a single investigator (A.D), beforehand calibrated with two experts in oral mucosa pathology (S.C, S.L) before the launch of the study by assessing 50 randomly selected images. The overall inter-rater kappa reached 0.92. Intra-observer reliability was tested by randomly re-examining 30 lesions at a one-month interval of the initial assessment and reached 0.87.

The diagnoses associated with each of the images were previously validated histologically. Anatomopathological analysis of oral lesions as part of the patient’s routine care was performed in the anatomopathological laboratory of the Toulouse University Hospital (CHU de Toulouse) and the Toulouse University Cancer Institute (IUCT, Institut Universitaire du Cancer de Toulouse). Demographic (age, gender, alcohol consumption, smoking status), medical data and medications were also recorded. Medical history and medications were, respectively, classified according to the International Classification of Diseases (ICD) and the Anatomical Therapeutic Chemical Classification System (ATC).

### 2.3. Model Development and Data Analysis

The machine learning pipeline involved (1) an adequate pre-processing step of the database (data encoding, imputing, and normalization), (2) a random train:test dataset splitting of 80:20, (3) the comparison and choice of a suitable predictive model, and (4) the explainability of the prediction results.

**Pre-processing step:** The binary categorical data (yes/no or male/female) were first label-encoded by integers. The type of elementary lesion was one hot encoded for non-tree-based ML models. Missing socio-demographic and/or medical data were considered as missing at-random, for age (68, 7.1%), gender (61, 6.4%), tobacco (7, 0.7%), alcohol (7, 0.7%) and narcotics (7, 0.7%). Missing data were considered as such for the algorithmic approach being able to deal with (tree-based gradient boosting method LightGBM) or imputed using a multivariate single imputation model [23].

**Data modeling:** Different Machine Learning strategies were compared: LightGBM (LGB, a tree-based gradient boosting method), Logistic Regression with an elastic net (L1 and L2) penalization (LR), Decision tree (DT), K-Nearest Neighbors (KNN) and Multi-Layer Perceptron (MLP). Models were used from scikit-learn 1.0.1 and lightgbm 3.3.1. To optimize the hyperparameters of each model, a grid search was explored using optuna 2.10.0, with mean ROC AUC one-versus-one after 5-fold cross-validation considered as eval metric.

**Model performance:** For each model, were computed the true positive rate (the proportion of positive data that are correctly considered as positive), the true negative rate (proportion of negative data that are correctly considered as negative), the positive predictive value (number of correct positive results divided by the number of positive results predicted by the classifier), the negative predictive value (number of correct negative results divided by the number of negative results predicted by the classifier), the overall accuracy (the number of correct predictions divided by the total number of predictions) and the F1-score (the harmonic mean between predictive positive values and true positive rate).

**Model explainability:** Finally, the influence of each attribute of the dataset on the diagnosis prediction was explored using SHapley Additive exPlanations (SHAP) [24]. Given the high dimensionality of the dataset (high number of variables) and the nature of the final chosen model (tree-based model lightGBM), the model was explained thanks to the TreeSHAP optimization [25].

## 3. Results

The dataset consisted of a total of 1242 records representing 31 different diagnoses for 299 subjects (39.4% men; 60.6% women; mean aged 57.3 years ± 19.2). Given the number of cases by diagnosis, 7 categories were considered: oral lichen planus (40.0% of the dataset), bullous diseases (13.9%), aphthous ulcers (8.9%), leukoplakia (7.3%), gingival enlargement (3.7%), oral squamous cell carcinoma (3.3%), and others (22.9%). A detailed description of the final dataset (socio-demographic and medical data) is provided in Table 1. Only 8 (2.7%) subjects were alcohol-dependent and 72 (24.1%) were smokers. The most represented elementary lesions were plaque/papule (27%), with 36% of these lesions being located on the attached gingiva.

Once models were tuned for optimization of hyperparameters and trained on the training dataset, the performance of each model was shown on the test dataset. LightGBM model achieved the best performance with 0.84 for mean accuracy. The other ML models achieved lower performances with 0.54, 0.73, 0.49, and 0.62, respectively, for logistic regression (LR), k-nearest neighbors (K-NN), decision tree classifier (DTC), and multilayer perceptron models (MLP). The detailed performances of the models are exposed in Table 2. Confusion matrixes are exposed in Figure 1A and Appendix A for LightGBM and other models, respectively.

Whatever the diagnosis to be predicted and the metrics, LightGBM achieved the best performances. The TreeSHAP method was used to provide insight into the explainability of the predictions. Figure 1B shows the importance of the different features in overall decision-making. The factors influencing the overall decision making are essentially lesion characteristics (type of lesion, color, reticulation…). Although age remains the most important factor in decision-making, other socio-demographic data are considered but with less importance. The model does not take into account the medical history and treatments in the overall explanation of the model behavior. Figure 2 and Appendix A expose the importance of the different features for the prediction of each diagnosis. If patients’ age seems to influence all different lesions, for Oral Lichen planus features such as the reticular form of the lesion, gender, type of elementary lesion, and localization have more influence on the diagnosis than the patient’s medical history. On the contrary, for OSSC, the size of the lesion, its localization, and the patient’s smoking status are more important to predict the diagnosis than the color of the lesion or the gender. Medical history often has a minor influence on individual prediction, with the exception of a history of dermatologic pathology, which strongly influences the diagnosis of bullous disease.

Finally, Figure 3 illustrates two examples of prediction, respectively, for a carcinoma (Figure 3A) and an oral lichen planus lesion (Figure 3B). The SHAP algorithm provides an understanding of decision making by outlining the factors that positively or negatively influence the different diagnoses.

## 4. Discussion

Based on the lesion topography and medical and socio-demographic patients features, LightGBM model allowed a prediction of the 7 oral mucosa lesions selected in this study with an overall accuracy of approximately 0.84. The specificity (true negative rate) and sensitivity (true positive rate) remain very high, ranging from 0.89 to 1.00 and 0.72 to 0.92, respectively; thus, illustrating the discriminatory capacity of our model. Therefore, this tool seems suitable for non-specialist practitioners to guide them in their diagnostic and therapeutic process.

The early diagnosis of oral lesions presents a challenge in treating them promptly and reducing therapeutic penalties, particularly in the case of cancer. Especially because therapeutic progress has increased the life expectancy of these cancer patients. Although oral dermatology is at the crossroads of several medical specialties (Dentists, Ear-Nose-Throat specialists, Oral and maxillofacial surgeons, dermatologists), oral lesions remain underdiagnosed, especially for precancerous and cancerous lesions [26,27]. Indeed, general practitioners often lack specific training and patients face a lack of referral to specialists in the field [28]. On the other hand, while the contribution of AI is booming in medicine and especially in dermatology, only a few models have been developed in oral dermatology [4,8,29]. To answer this problem, we have demonstrated that we can accurately predict the diagnosis of an oral mucosal lesion using an algorithmic ML strategy based on routine oral medicine consultation. Similarly, other models created for the automatic detection of skin cancers have shown the importance of including patient clinical information to optimize these tools [30].

Machine learning algorithms are considered as “black boxes”. Explainability aims at highlighting the importance of each data in the decision-making process to improve transparency, questioning, and understanding.

This explanability increases the confidence of the models by ensuring that the decision making is coherent with the semiological data classically accepted in the literature. Furthermore, the identification of new correlations must be supported by evidence to eliminate misleading decisions. Finally, understanding the models is not only a research objective; it allows the practitioner to be guided in his clinical thinking, to identify variables of high importance and, therefore, optimize the diagnosis of lesions. Moreover, while the explainability of a model can highlight important interactions between certain data, it does not allow us to conclude that there are causal links.

The development of applications accessible to the general public for preventive and diagnostic purposes is an important issue [31]. In 2018, the systematic review by Chuchu et al. [32] evaluated the sensitivity and specificity of smartphone apps used for the diagnosis of melanoma. Although promising, the results showed a lack of precision that led to diagnostic errors compared to the diagnosis made by expert dermatologists. Two years later, Udréa et al. in 2020 [33] improved the system by developing a smartphone application based on a machine learning algorithm that allows the detection of cancerous or precancerous skin lesions with a sensitivity of 95.1% and a specificity of 77.3%. In the diagnosis of oral lesions, by allowing the detection of “high risk cancer patients” using patient’s socio-demographic and behavior data associated with elementary lesion description, the present algorithm can thus constitute a simple tool for specialists and non-specialists to use, with a high precision accuracy.

Today, the most powerful AI models used for diagnosis in dermatology are now based on deep convolutional neural networks. They allow the image of the lesion to be used considering the pixels that constitute it as features in their own right [4,34,35,36], even associating information about the patient [37]. The use of the photograph itself to determine the diagnosis is the logical continuation of our work, especially to refine the prediction.

Based on the results of such study, our algorithm could be used in routine practice to assist in diagnosis and perform a first triage. In fact, it could make it easier to detect patients with lesions at risk of malignant development. Of course, it does not allow one to date to replace the specialist due to some limitations.

Some issues are currently being raised to train machine learning models. Unlike skin lesions, there is currently no public database available for oral lesions [38,39]. There is also a very important clinical polymorphism of certain oral mucosa lesions, which can complicate the prediction. Finally, as endo-buccal photos are difficult to standardize, it will be necessary to normalize the images as much as possible to compare them [40]. This is an essential step in the development and generalization of this device to predict not only cancerous or precancerous lesions, but also other often misdiagnosed oral mucosal lesions that also have an impact on patient quality of life and well-being.

A limitation of the present study is the limited size of the database and the imbalance between the different diagnoses chosen to predict. Even if this has been taken into account and compensated for in the algorithm, it would be preferable to train it on a larger scale with more cases and a more balanced dataset. However, to date there is no international database on oral mucosal lesions, unlike databases on skin lesions.

## 5. Conclusions

The use of ML models in medicine is promising, especially due to their ability to compute a very large amount of data to make predictions. Although there are some limitations in our database (size of the database, picture standardization), the results of our study are promising in that they show a high accuracy for predicting cancerous lesions (0.90 accuracy) and an overall good accuracy of 0.84 for five of the other most frequent oral mucosal lesions (Oral Lichen planus, Leukoplakia, bullous diseases, aphthous ulcers, gingival enlargements). More studies with much larger databases should be carried out to validate the accuracy of our model. Moreover, the use of deep learning models could be particularly interesting and should be investigated to optimize diagnostic devices to predict directly from the image of the lesion. These tools create new opportunities for democratization and improvement of patient care. However, the implementation of AI-based applications raises other questions about issues such as the acceptability of AI in health or data protection. 

## Figures and Tables

**Figure 1 jcm-11-06596-f001:**
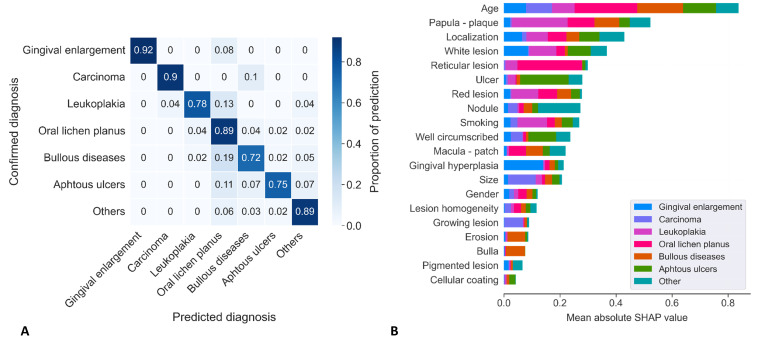
Confusion matrix for LightGBM model and Feature importance for overall predictions. (**A**) The confusion matrix describes the performance of the LightGBM classifier on the test dataset. The number of correctly classified lesions ranged from 92% to 72% depending on the different diagnoses evaluated. (**B**) SHAP feature importance for the LightGBM trained model for predicting oral mucosa lesions measured as the mean absolute SHAP values. Abbreviations: ATC A, Anatomical Therapeutic Chemical classification: Alimentary tract and metabolism; ATC C, Anatomical Therapeutic Chemical classification: Cardiovascular system; ICD E00-E89, in-ternational classification of diseases: Endocrine, nutritional and metabolic diseases; ICD I00-I99, international classification of diseases: Diseases of the circulatory system; ICD K00-K95, international classification of diseases: Diseases of the digestive system; ICD L00-L99, international classification of diseases: Diseases of the skin and subcutaneous tissue; ICD M00-M99, international classification of diseases: Diseases of the musculoskeletal system and connective tissue.

**Figure 2 jcm-11-06596-f002:**
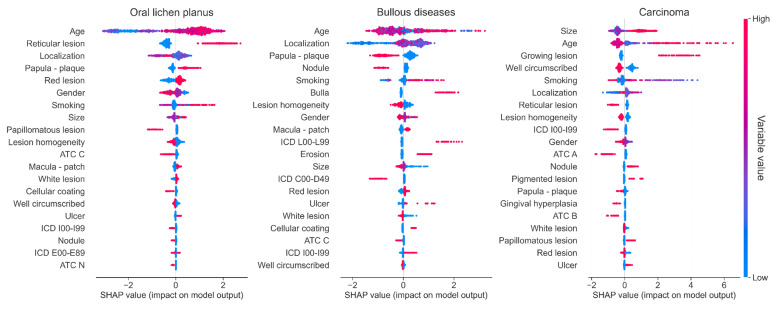
Summary plot of feature importance for each diagnostic prediction for Oral lichen planus, Bullous diseases and Carcinoma. Each point on the summary plot is a SHAP value for a given feature and instance. The position on the y-axis is determined by the feature and on the *x*-axis by the SHAP value. The color represents the value of the feature from low to high. Overlapping points are jittered in *y*-axis direction, so we get a sense of the distribution of the SHAP values per feature. The features are ordered according to their importance. Abbreviations: ATC A, Anatomical Therapeutic Chemical classification: Alimentary tract and metabolism; ATC C, Anatomical Therapeutic Chemical classification: Cardiovascular system; ATC H, Anatomical Therapeutic Chemical classification: Systemic hormonal preparations; ATC M, Anatomical Therapeutic Chemical classification: Musculo-skeletal system; ATC N, Anatomical Therapeutic Chemical classification: Nervous system; ATC R, Anatomical Therapeutic Chemical classification: Respiratory system; ICD C00-D49, international classification of diseases: Neoplasms; ICD E00-E89, international classification of diseases: Endocrine, nutritional and metabolic diseases; ICD F00-F99, international classification of diseases: Mental and behavioral disorders; ICD G00-G99, international classification of diseases: Diseases of the nervous system; ICD I00-I99, international classification of diseases: Diseases of the circulatory system; ICD K00-K95, international classification of diseases: Diseases of the digestive system; ICD L00-L99, international classification of diseases: Diseases of the skin and subcutaneous tissue; ICD M00-M99, international classification of diseases: Diseases of the musculoskeletal system and connective tissue; ICD N00-N99, international classification of diseases: Diseases of the genitourinary system.

**Figure 3 jcm-11-06596-f003:**
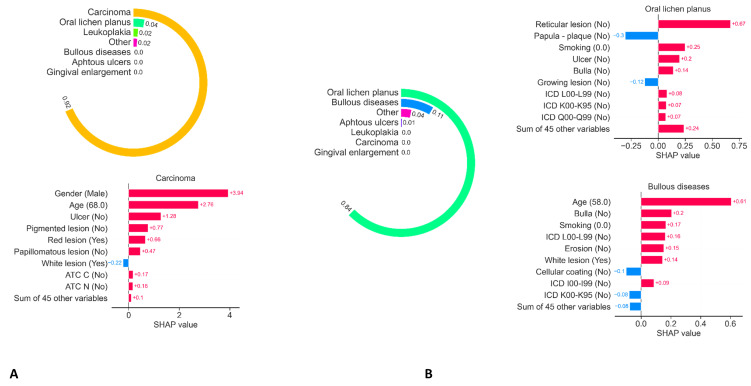
Illustration of two examples of prediction, respectively for a carcinoma (**A**) and an oral lichen planus lesion (**B**). The SHAP algorithm provides an understanding of decision making by outlining the factors that positively or negatively influence the different diagnoses. Abbreviations: ATC A, Anatomical Therapeutic Chemical classification: Alimentary tract and metabolism; ATC C, Anatomical Therapeutic Chemical classification: Cardiovascular system; ATC H, Anatomical Therapeutic Chemical classification: Systemic hormonal preparations; ATC M, Anatomical Therapeutic Chemical classification: Musculo-skeletal system; ATC N, Anatomical Therapeutic Chemical classification: Nervous system; ATC R, Anatomical Therapeutic Chemical classification: Respiratory system; ICD C00-D49, international classification of diseases: Neoplasms; ICD E00-E89, international classification of diseases: Endocrine, nutritional and metabolic diseases; ICD F00-F99, international classification of diseases: Mental and behavioral disorders; ICD G00-G99, international classification of diseases: Diseases of the nervous system; ICD I00-I99, international classification of diseases: Diseases of the circulatory system; ICD K00-K95, international classification of diseases: Diseases of the digestive system; ICD L00-L99, international classification of diseases: Diseases of the skin and subcutaneous tissue; ICD M00-M99, international classification of diseases: Diseases of the musculoskeletal system and connective tissue; ICD N00-N99, international classification of diseases: Diseases of the genitourinary system.

**Table 1 jcm-11-06596-t001:** Subject characteristics are used for machine learning model training according to each diagnosis.

	Lichen Planus	Leukoplakia	Aphthous Ulcer	Bullous Diseases	Gingival Enlargement	Carcinoma	Others	*p*-Value ^a^
Number of Subjects	(n = 77)	(n = 41)	(n = 37)	(n = 26)	(n = 12)	(n = 11)	(n = 95)	
**Gender (n,%)**								0.03
Male	22 (29%)	20 (49%)	19 (51%)	9 (35%)	5 (42%)	5 (45%)	42 (44%)	
Female	51 (66%)	16 (39%)	14 (38%)	15 (58%)	3 (25%)	4 (36%)	53 (56%)	
Mean age	65	65	50	65	43	54	50	<0.001
Alcohol (n,%)	2 (3%)	4 (10%)	1 (3%)	-	-	2 (18%)	2 (2%)	0.007
Smoking (n,%)	21 (27%)	18 (44%)	8 (22%)	3 (12%)	1 (8%)	5 (45%)	16 (17%)	0.002
**Number of lesions (n)**	497	91	110	173	46	41	252	
**Localization (n,%)**								0.002
Vermilion	19 (3.8 %)	6 (6.5%)	5 (4.5%)	3 (1.7%)	-	-	16 (6.3%)	
Commissure	1 (0.2%)	-	-	1 (9.6%)	-	-	8 (3%)	
Labial mucosa	8 (1.7%)	3 (3%)	15 (13.6%)	16 (9.2%)	-	1 (2.4%)	21 (8.3%)	
Muco-buccal fold	136 (27.3%)	25 (27.5%)	18 (16.4%)	16 (9.2%)	-	5 (12%)	21 (8.3%)	
Attached gingiva	172 (34.6%)	12 (13.1%)	12 (10.9%)	91 (52.6%)	45 (98%)	16 (39%)	85 (33.7%)	
Buccal mucosa	37 (7.4%)	3 (3.3%)	10 (9.0%)	14 (8.0%)	1 (2.0%)	2 (4.8%)	11 (4.4%)	
Hard palate	22 (4.4%)	13 (14.3%)	5 (4.5%)	10 (5.7%)	-	11 (26.8%)	20 (7.9%)	
Soft palate	1 (0.2%)	4 (4.4%)	5 (4.5%)	19 (10.9%)	-	-	2 (0.7%)	
Tonsillar pillar	3 (0.6%)	1 (1.1%)	1 (0.9%)	1 (0.5%)	-	-	2 (0.7%)	
Dorsal tongue	60 (12.0%)	7 (1.8%)	3 (2.7%)	2 (1.2%)	-	2 (4.8%)	51 (20.2%)	
Ventral tongue	9 (1.8%)	1 (1.0%)	10 (9.0%)	6 (3.5%)	-	1 (2.4%)	2 (0.7%)	
Lateral tongue	24 (4.8%)	14 (15.4%)	13 (11.8%)	-	-	1 (2.4%)	11 (4.4%)	
Floor of mouth	5 (1.0%)	2 (2.2%)	2 (1.8%)	-	-	2 (4.8%)	2 (0.7%)	
**Lesion color (n,%)**								<0.001
White	160 (32.0%)	70 (7.8%)	33 (30.0%)	46 (26.6%)	-	13 (31.7%)	50 (18.8%)	
Red	55 (11.0%)	2 (2.2%)	9 (8.1%)	55 (31.8%)	15 (32,6%)	8 (19.5%)	51 (20.2%)	
Mixt (red and white)	268 (54%)	19 (20.8%)	66 (60.0%)	69 (39.9%)	5 (11.8%)	16 (39.0%)	79 (31.3%)	
Pigmented	10 (2.0%)	-	1 (0.9%)	-	5 (11.8%)	3 (2.3%)	40 (15.8%)	
No color changes	4 (0.8%)	-	1 (0.9%)	-	21 (46.0%)	1 (2.4%)	32 (12.7%)	
**Lesion size (n,%)**								<0.001
<5 mm	39 (7.8%)	13 (14.3%)	27 (24.5%)	32 (18.5%)	27 (58.6%)	1 (2.4%)	41 (16.2%)	
5 to 10 mm	268 (54.0%)	41 (45.1%)	66 (60.0%)	116 (67.0%)	38 (82.6%)	9 (21.9%)	135 (53.1%)	
10 to 50 mm	188 (38.0%)	37 (40.7%)	17 (15.5%)	25 (14.5%)	6 (13.0%)	27 (65.8%)	75 (29.7%)	
>50 mm	2 (0.4%)	-	-	-	-	4 (9.7%)	1 (0.4%)	
**Elementary Lesion (n,%)**								<0.001
Gingival hyperplasia	3 (0.6%)	-	-	1 (0.5%)	40 (86.9%)	-	9 (3.5%)	
Bulla	7 (1.4%)	2 (2.2%)	2 (1.8%)	31 (18.0%)	-	-	16 (6.3%)	
Cellular coating	11 (2.2%)	1 (1.1%)	13 (11.8%)	16 (9.2%)	-	2 (4.8%)	12 (4.7%)	
Erosion	18 (3.6%)	3 (3.3%)	6 (5.5%)	30 (18.3%)	-	1 (2.4%)	8 (3.1%)	
Macula/patch	184 (37.0%)	12 (13.2%)	11 (10.0%)	79 (45.7%)	6 (13.0%)	4 (9.7%)	78 (30.9%)	
Nodule	13 (2.6%)	2 (2.2%)	2 (1.8%)	-	-	18 (43.9%)	100 (39.6%)	
Papula/plaque	220 (44.3%)	70 (77.0%)	8 (7.3%)	7 (4.0%)	-	13 (31.7%)	23 (9.1%)	
Ulcer	40 (8.0%)	1 (1.1%)	68 (61.8%)	9 (5.2%)	-	3 (7.3%)	4 (1.5%)	
Lesion homogeneity (n,%)	187 (37.6%)	52 (75.1%)	77 (70.0%)	71 (41.0%)	32 (69.5%)	5 (12.1%)	142 (56.4%)	<0.001
Well circumscribed lesion (n,%)	230 (46.3%)	55 (60.4%)	93 (84.5%)	81 (46.8%)	19 (41.3%)	7 (17.0%)	166 (65.8%)	<0.001
Elevated edges of the lesion (n,%)	5 (1.0%)	-	13 (11.8%)	2 (1.2%)	1 (2.0%)	1 (2.4%)	4 (1.6%)	0.29
Reticular lesion (n,%)	175 (35.2%)	3 (3.3%)	1 (0.9%)	5 (2.9%)	-	-	5 (1.9%)	<0.001
Growing lesion (n,%)	13 (2.6%)	2 (2.2%)	-	1 (0.5%)	3 (6.5%)	24 (58.5%)	25 (9.9%)	<0.001
Papillomatus lesion (n,%)	3 (0.6%)	8 (8.8%)	1 (0.9%)	1 (0.5%)	1 (2.0%)	8 (19.5%)	41 (16.2%)	0.007

^a^ for categorical values, the *p*-value resulted from a χ^2^ test; for numerical values, the *p*-value resulted from an ANOVA test.

**Table 2 jcm-11-06596-t002:** Models’ performances for each diagnosis.

	LightGBM					Elastic Net Regression					K-Nearest Neighbors					Decision Tree					Multilayer Perceptron				
**Diagnostics**	**TPR**	**TNR**	**PPV**	**NPV**	**F1**	**TPR**	**TNR**	**PPV**	**NPV**	**F1**	**TPR**	**TNR**	**PPV**	**NPV**	**F1**	**TPR**	**TNR**	**PPV**	**NPV**	**F1**	**TPR**	**TNR**	**PPV**	**NPV**	**F1**
Gingival enlargement	0.92	1.00	1.00	1.00	0.96	0.92	0.99	0.85	1.00	0.88	0.83	0.99	0.71	0.99	0.77	0.67	0.97	0.47	0.99	0.55	0.83	1.00	1.00	0.99	0.91
Carcinoma	0.90	1.00	0.90	1.00	0.90	0.60	0.92	0.21	0.99	0.32	0.70	0.99	0.64	0.99	0.67	0.80	0.96	0.40	0.99	0.53	0.50	0.98	0.50	0.98	0.50
Leukoplakia	0.78	0.98	0.75	0.98	0.77	0.74	0.91	0.40	0.98	0.52	0.57	0.94	0.42	0.96	0.48	0.74	0.89	0.36	0.98	0.49	0.35	0.98	0.62	0.95	0.44
Oral lichen planus	0.89	0.89	0.85	0.92	0.87	0.38	0.92	0.77	0.68	0.51	0.85	0.88	0.83	0.89	0.84	0.27	0.98	0.92	0.66	0.42	0.84	0.70	0.66	0.86	0.74
Bullous diseases	0.72	0.96	0.76	0.95	0.74	0.70	0.81	0.38	0.94	0.49	0.70	0.96	0.75	0.95	0.72	0.74	0.76	0.34	0.95	0.46	0.35	0.95	0.56	0.90	0.43
Aphthous ulcers	0.75	0.99	0.84	0.97	0.79	0.54	0.96	0.56	0.95	0.55	0.46	0.97	0.59	0.95	0.52	0.75	0.92	0.50	0.97	0.60	0.50	0.98	0.74	0.95	0.60
Other lesions	0.89	0.97	0.89	0.97	0.89	0.60	0.94	0.72	0.90	0.66	0.67	0.93	0.72	0.91	0.69	0.46	0.93	0.64	0.87	0.54	0.73	0.92	0.70	0.93	0.71
Mean accuracy	0.84					0.54					0.73					0.49					0.67				

Abbreviations: TPR, true positive rate; TNR, true negative rate; PPV, positive predictive value; NPV, negative predictive value. True positive rate: proportion of positive data that are correctly considered as positive. True negative rate: proportion of negative data that are correctly considered as negative. Positive predictive value: number of correct positive results divided by the number of positive results predicted by the classifier. Negative predictive value: number of correct negative results divided by the number of negative results predicted by the classifier. F1-score: the harmonic mean between predictive positive values and true positive rate.

## Data Availability

Data available on request from the authors.

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
