# Peer review of "Improvement of Mucosal Lesion Diagnosis with Machine Learning Based on Medical and Semiological Data: An Observational Study"

_jcm, 2022, doi:10.3390/jcm11216596_

Round 1

Reviewer 1 Report

I have read with great interest the manuscript entitled: "Oral Mucosal lesion diagnosis from patient data collection using Machine Learning".

The authors should respond to the following points:

TITLE:

- Authors should modify the title. The title of their manuscript should be more precise about the subject matter of the study.

ABSTRACT:

- Authors should indicate in the abstract the dates of their study. 

MATERIAL AND METHODS:

- Have the authors obtained a favourable report from a Bioethics Commission?

- What were the inclusion and exclusion criteria for this study?

- In which centre was the anatomopathological analysis of the lesions performed?

- What was the age range of the study participants?

RESULTS:

- The results are described in a clear and concise manner.

DISCUSSION:

- What is the practical application of this study?

CONCLUSIONS:

- The conclusions should stick to the results obtained in the study. The authors should modify this point based on their findings.

BIBLIOGRAPHY:

- Bibliographical references are not described according to journal guidelines.

Author Response

Toulouse, October 09, 2022

“Improvement of mucosal lesion diagnostic with Machine Learning based on medical and semiological data : An observational study”

Manuscript ID: JCM-1966994

***************************************************************

We would like to thank the reviewers for the time they spent on their careful examination of our manuscript and their very constructive comments and suggestions. 

Our responses to the reviewers are noted point by point below. We give our explanations in response to each comment below, together with the changes made. These changes are printed in red, both here and in the manuscript.

We hope that the modifications made will be satisfactory and will enhance the quality of the manuscript.

Reviewer #1

1- TITLE:

- Authors should modify the title. The title of their manuscript should be more precise about the subject matter of the study.

Improvement of mucosal lesion diagnosis with Machine Learning based on medical and semiological data : An observational study

2- ABSTRACT:

- Authors should indicate in the abstract the dates of their study. 

The changes were made as follows : 

This cohort study included patients followed between January 2015 and December 2020 in the oral mucosal pathology consultation of Toulouse university hospital

3- MATERIAL AND METHODS:

- Have the authors obtained a favourable report from a Bioethics Commission?

As indicated in the French legislation and in the French National Commission for Informatics and Liberties (CNIL). The study being retrospective, and the consent of each patient having been obtained, the authors did not have to ask the opinion of a bioethics committee.

- What were the inclusion and exclusion criteria for this study?

The authors apologize for the lack of precision and all the inclusion/exclusion criteria were added in the manuscript : 

All patients meeting the following criteria were included in the study:

- Patients consulting in the oral mucosa pathology consultation of the Toulouse University Hospital and presenting a mucosal lesion.

- Patients already followed for such lesions and requiring a clinical and/or histological re-evaluation

- Subjects older than 18 years of age at the time of consultation

- Subjects who are able to receive information and understand the information about the study. This implies: to master the French language and not be subject to a restriction of rights by the judicial authorities

- subjects who have given their consent to participate in the study.

Patients with the following conditions could not be included in this study:

- Pregnant and lactating women

- Patients under guardianship, curatorship or safeguard of justice

- In which center was the anatomopathological analysis of the lesions performed?

Precisions have been made as follows : 

Anatomopathological analysis of oral lesions as part of the patient's routine care was performed in the anatomopathological laboratory of the Toulouse University Hospital (CHU de Toulouse) and the Toulouse University Cancer Institute (IUCT, Institut Universitaire du Cancer de Toulouse).

- What was the age range of the study participants?

  The authors have indeed lacked clarity on this point.  The Age range has been added to the text.

4- DISCUSSION:

- What is the practical application of this study?

Based on the results of our study, our algorithm could be used in routine practice to assist in the diagnosis and perform a first triage. Indeed, it could make it easier to detect patients with lesions at risk of malignant development. Earlier diagnosis of these lesions could help limit patient wandering and improve patient care. However, it is not able, to date, to replace the specialist because of some limitations.

5- CONCLUSIONS:

- The conclusions should stick to the results obtained in the study. The authors should modify this point based on their findings.

We thank you for the remark and modified the conclusion according to the reviewers suggestion.

The use of ML models in medicine is promising, especially through their ability to compute a very large amount of data to make predictions. Even though there are some limitations in our database (size of the database, picture standardization), the results of our study are promising in that they show a high accuracy for predicting cancerous lesions (0.90 accuracy) and  an overall good accuracy of 0.84  for five of the other most frequent oral mucosal lesions (Oral Lichen planus, Leukoplakia, bullous diseases, aphthous ulcers, gingival enlargements). Further studies with much larger databases should be carried to validate the accuracy of our model. Moreover, the use of deep learning models could be particularly interesting and should be investigated to optimize diagnostic devices in order to predict directly from the image of the lesion. These tools create new opportunities for democratization and improvement of patient care. However, the implementation of applications based on AI raises other questions about issues such as the acceptability of AI in health or data protection. 

6- BIBLIOGRAPHY:

- Bibliographical references are not described according to journal guidelines.

We thank the reviewers for this remark, the references style has been modified accordingly to match the expectations of the journal.

Reviewer 2 Report

The authors developed and evaluated a machine learning algorithm to predict oral mucosa lesions diagnosis. They used data based on chair-side patient information collection such as semiology of the lesions, patient socio-demographic and medical data, for diagnosis of oral mucosa lesions.

The data set is sufficiently large (The dataset consisted of a total of 2,496 records). LightGBM is a relatively new algorithm, a gradient boosting framework that uses a tree-based learning algorithm. Light GBM can handle the large size of data and takes lower memory to run. Another reason why Light GBM is popular is that it focuses on the accuracy of results. The paper is original and well written, with only minor issues in the English language.

Author Response

Toulouse, October 09, 2022

“Improvement of mucosal lesion diagnostic with Machine Learning based on medical and semiological data : An observational study”

Manuscript ID: JCM-1966994

***************************************************************

We would like to thank the reviewers for the time they spent on their careful examination of our manuscript and their very constructive comments and suggestions. 

Our responses to the reviewers are noted point by point below. We give our explanations in response to each comment below, together with the changes made. These changes are printed in red, both here and in the manuscript.

We hope that the modifications made will be satisfactory and will enhance the quality of the manuscript.

Reviewer #2

The authors thank you for your comments and consideration.

Round 2

Reviewer 1 Report

Authors should pay attention to the following points in their manuscript:

1) In line 77 it is more precise to indicate that patients of legal age were included in the study, patients over 18 years of age. 

2) In the manuscript, the authors should indicate that according to French legislation and as it is a retrospective study, as the authors themselves say, it is not necessary to request authorization from a Bioethics Committee. This point should be justified by means of the corresponding bibliographic reference. The authors should indicate in the text of the manuscript why they have not requested the authorization of a Bioethics Committee.

3) The authors have not corrected the bibliographic references. The bibliographic references are still not described according to the regulations of the journal. Why do they not write the title of the journals? 

Authors should pay special attention to this point. 

Author Response

Toulouse, November 1, 2022,

“Improvement of mucosal lesion diagnostic with Machine Learning based on medical and semiological data : An observational study”

Manuscript ID: JCM-1966994

***************************************************************

We would like to thank the reviewers for the time they spent on their careful examination of our manuscript and their very constructive comments and suggestions.

Our responses to the reviewers are noted point by point below. We give our explanations in response to each comment below, together with the changes made. These changes are printed in red, both here and in the manuscript.

We hope that the modifications made will be satisfactory and will enhance the quality of the manuscript.

Reviewer #1

  • In line 77 it is more precise to indicate that patients of legal age were included in the study, patients over 18 years of age.

The authors have modified this point in accordance with your remarks

  • In the manuscript, the authors should indicate that according to French legislation and as it is a retrospective study, as the authors themselves say, it is not necessary to request authorization from a Bioethics Committee. This point should be justified by means of the corresponding bibliographic reference. The authors should indicate in the text of the manuscript why they have not requested the authorization of a Bioethics Committee.

These details were added in the last revision. However, for the sake of clarity we have adapted the text. You will also find the necessary reference.

Please find attached the link to the French law (https://www.legifrance.gouv.fr/codes/article_lc/LEGIARTI000046125746)

  • The authors have not corrected the bibliographic references. The bibliographic references are still not described according to the regulations of the journal. Why do they not write the title of the journals?

Authors should pay special attention to this point.

The authors apologize for this error. We have modified the format of the references according to the mdpi guidelines.